# RNA Sequencing-Based Whole-Transcriptome Analysis of Friesian Cattle Fed with Grape Pomace-Supplemented Diet

**DOI:** 10.3390/ani8110188

**Published:** 2018-10-23

**Authors:** Marco Iannaccone, Ramy Elgendy, Mery Giantin, Camillo Martino, Daniele Giansante, Andrea Ianni, Mauro Dacasto, Giuseppe Martino

**Affiliations:** 1Faculty of Bioscience and Technology for Food, Agriculture, and Environment, University of Teramo, Via R. Balzarini 1, 64100 Teramo, Italy; m.iannaccone@unina.it (M.I.); andreaianni@hotmail.it (A.I.); 2Department of Comparative Biomedicine and Food Science, University of Padua, viale dell’Università 16, 35020 Legnaro (Padua), Italy; ramy.elgendy@igp.uu.se (R.E.); mery.giantin@unipd.it (M.G.); mauro.dacasto@unipd.it (M.D.); 3Department of Immunology, Genetics and Pathology, Uppsala University, 75185 Uppsala, Sweden; 4Department of Veterinary Medicine, University of Perugia, Via S. Costanzo 4, 06126 Perugia, Italy; camillo.martino91@gmail.com; 5Istituto Zooprofilattico Sperimentale dell’Abruzzo e del Molise “G. Caporale”, Campo Boario, 64100 Teramo, Italia; d.giansante@izs.it

**Keywords:** grape pomace, transcriptomics, RNA-seq, cholesterol biosynthesis, blood, cattle

## Abstract

**Simple Summary:**

Grape pomace (GPO) is an important source of polyphenols which are known to have antioxidant properties. In the past decade, GPO has received some attention as a bioactive dietary component in farm animals’ diet. In this study, we have analyzed the whole-transcriptome of Friesian calves fed with a GPO-supplemented diet using RNA-sequencing. We noted that the most affected pathway was the cholesterol lipid biosynthesis and this effect was consistent with a reduction in both serum cholesterol and lipid oxidation in the carcasses. This study provides evidence on the antioxidant property of GPO-supplemented diet, from a molecular biology standpoint.

**Abstract:**

Grape pomace (GPO), the main by-product of the wine making process, is a rich source of polyphenols with potent antioxidant properties. Recently, GPO has emerged as a potential feed additive in livestock nutrition, with several reports describing its beneficial effects on animals’ overall health status or production traits. However, little is known about it from a molecular biology standpoint. In the present study, we report the first RNA sequencing-based whole-transcriptome profiling of Friesian calves fed with a GPO-supplemented diet. We identified 367 differentially expressed genes (*p* < 0.05) in the GPO-supplemented calves (*n* = 5), when compared with unsupplemented control group (*n* = 5). The pathway analysis showed that ‘cholesterol lipid biosynthesis’ was the most negatively-enriched (*p* < 0.001) pathway in the GPO-supplemented animals. In specific terms, five important genes coding for cholesterol biosynthesis enzymes, namely the Farnesyl-diphosphate Farnesyltransferase 1 (FDFT-1), Squalene Epoxidase (SQLE), NAD(P)-dependent Steroid Dehydrogenase-like (NSDHL), Methylsterol Monooxygenase (MSMO)-1, and Sterol-C5-desaturase (SC5D), two major transcription factors (the Sterol Regulatory Element-binding Transcription Factor 1 and 2), as well as the Low-Density Lipoprotein Receptor (LDLR), were all downregulated following GPO supplementation. Such an effect was mirrored by a reduction of blood cholesterol levels (*p* = 0.07) and a lowered (*p* < 0.001) Malondialdehyde (lipid oxidation marker) level in carcasses. We provide evidence on the effects of GPO-supplemented diets on the whole-transcriptome signature in veal calves, which mainly reflects an antioxidant activity.

## 1. Introduction

The inclusion of mineral elements or some agro-industrial by-products in animals’ diet is becoming a desirable practice in livestock production due to its beneficial effects on animals’ production and overall health [1,2]. Grape pomace (GPO) is the solid by-product of the winemaking process and, when left unused, represents an important environmental disposal issue. For this reason, as well as for its possible nutritive values, GPO has been proposed as a diet supplement in farm animals and humans [3,4]. Indeed, chemical constituent analyses revealed that GPO is rich in polyphenols, including phenolic acid, flavonoids, procyanidins, and anthocyanins, whose biological functions have been extensively studied. For example, in broilers where the meat is rich in polyunsaturated fatty acid, a GPO-supplemented diet improved meat quality delaying lipid oxidation and reducing the potential risk of secondary product formation [5]. In pigs, GPO showed a beneficial effect on fatty acid composition, where it reduced palmitic, stearic, and arachidic acid levels, while increasing linoleic acid [6]. Moreover, when antioxidants from GPO were added to meat, the shelf life of the meat products was improved [7]. However, data about the use of GPO-supplemented diets in ruminants are controversial. In sheep, a high percentage of GPO reduced the dry matter apparent digestibility due to the GPO’s high content of lignin [8]. On the contrary, lower amounts of GPO showed a less negative effect on apparent digestibility together with a prevention of milk fatty acids oxidation in dairy cows [9]. Furthermore, the milk from dairy cows fed with a GPO-supplemented diet had an increased β-lactoglobulin protein fraction [10], that is believed to be associated with several biological activities including an immunomodulatory and hypocholesterolemic effects [11].

Diet constituents are known to affect the host’s transcriptome through the release of bioactive derivatives. Therefore, besides the classical evaluation of animal performance and productive traits, it is nowadays important to assess the molecular mechanisms and signaling and metabolic pathways surrounding animal feeds and feed additives [12]. While the ‘overall’ antioxidant effect of GPO supplementation seems to be well-documented in both small [13,14] and large ruminants [9,15], there is little information available on the molecular machinery through which GPO exert its effects. Our hypothesis was that dietary GPO would induce metabolism- and redox-related genes and leave a noticeable molecular signature behind. Therefore, in this study we sought a nutrigenomics-based approach, in which we profiled, by RNA-Seq, the whole-transcriptome of GPO-supplemented veal calves.

## 2. Materials and Methods

All the experimental procedures and the handling of animals mentioned in the present study were carried out according to the national legislation on animal welfare (DL n. 126, 7 July 2011). The experimental animals were hosted, managed, and supplemented in a commercial farm in the region of Abruzzo (Italy) that follows the national and European regulation regarding livestock (DL n. 126, 7 July 2011, EC Directive 51 2008/119/EC), and then slaughtered in compliance with the European Union regulation (EC 1099/2009) on the protection of animals at the time of killing.

### 2.1. Animals and Sampling

Ten male Friesian calves, homogenous for age and weight, were used in this study. The trial was conducted in a farm in the region of Abruzzo (Italy). Animals were randomly divided into two experimental groups of five calves each: a control group (CTR), and a group receiving a standard diet supplemented with dry GPO flour (group GPO). Calves had a 4-month (120-day) acclimatization period, in which they were kept on a basal diet that consisted mainly of alfalfa hay plus a custom-formulated concentrate offered to the animals *ad libitum* (Table 1)*.*

The acclimatization period was then followed by a 75-day supplementation period where the CTR group kept receiving the control diet, which consisted of mainly hay and a custom-formulated concentrate without GP meal and the GPO group received ad libitum the same diet plus concentrate which includes GP meal 10%. The composition of the GPO-supplemented and CRT-supplement is reported in Table 2.

At the end of the experiment the animals, ageing 8.3 ± 0.4 months and weighing 340 ± 35 kg, were slaughtered. Blood samples were collected in 10 mL Venoject glass tubes containing ethylenediaminetetraacetic acid (EDTA) (or sodium heparin; Terumo Italia, Rome, Italy), during the slaughter process, to measure plasma biochemical parameters. Tubes were immediately cooled and plasma was separated within 30 min of collection. After centrifugation at 500 RCF for 15 min, samples were stored at −20 °C until further analysis. For total RNA isolation, duplicate blood samples (2 × 2.5 mL/animal) were collected from the jugular vein in PAXgene RNA tubes (PreAnalytics/Qiagen, Milan, Italy) at two time-points, at the beginning (T0) and after 75 days of GPO supplementation (T75). Tubes were kept at room temperature overnight and, then stored at −20 °C until further processing.

### 2.2. Blood Analysis

The analyses were performed at the Veterinary and Public Health Institute, Teramo, Italy. The complete blood cell count with leukocyte formula (total white blood cells, monocyte, lymphocyte, basophils, neutrophils, and eosinophils) was performed using a laser-based hematology analyzer with software applications for animal species (ADVIA 120 hematology system, Siemens, Munich, Germany). Plasma samples were analyzed for different compounds (urea, calcium, tryglicerides, glucose, GOT, GPT) with an automatic biochemistry analyzer (ILAB 650, Instrumentation Laboratory-Werfen, Milan, Italy) and following the routine procedure of the institute (Veterinary and Public Health Institute “G. Caporale”, Teramo, Italy).

### 2.3. Lipid Oxidation Measurement

Twenty-four hours after slaughtering, meat samples were collected following the ASPA (Associazione Scientifica Produzioni Animali, Viterbo, Italy) guidelines and stored at −20 °C until analysis. To evaluate the effects of freezing and defrosting on lipid oxidation, we measured the 2-thiobarbituric acid reactive substances (TBARS) immediately after slaughtering (T0) and after 7 days (T7), following a detailed protocol previously reported [16].

### 2.4. RNA Isolation

Total RNA was isolated using the PAXgene blood RNA kit (PreAnalytics/Qiagen, Italy), as per the manufacturer’s instructions. To remove any genomic DNA traces, an in-column DNase treatment was performed for 15 min prior to RNA elution. RNA was eluted from the filter and stored at −80 °C until further processing. Total RNA concentration was determined using the NanoDrop ND-1000 UV-Vis spectrophotometer (NanoDrop Technologies Inc., Wilmington, DE, USA), and its quality was measured by the 2100 Bioanalyzer and RNA 6000 Nano kit (Agilent Technologies, Santa Clara, CA, USA). Only RNA samples with an RNA integrity number (RIN) ≥ 8 were selected for the RNA-Seq library preparation.

### 2.5. Library Preparation, Sequencing, and RNA-Seq Analyses

Using 0.5 µg of total RNA as input, poly(A) mRNA was enriched using the NEB magnetic mRNA isolation kit, and strand-specific RNA-seq libraries were prepared using the NEBNext Ultra RNA Library Prep Kit for Illumina (New England BioLabs, Ipswich, MA, USA) as per the manufacturer’s specifications. The Illumina-specific adaptor was sequentially ligated to the 3′ end of cDNA fragments, purified using the AMPure XP beads (Beckman Coulter, Brea, CA, USA) and, finally, PCR-amplified (14 cycles) using an appropriate indexing primer to allow further samples multiplexing. The PCR-amplified libraries were purified again by the AMPure XP beads (Beckman Coulter, Brea, CA, USA) and then assessed for their quality and fragments distribution using the 2100 Bioanalyzer DNA 1000 assay (Agilent Technologies, Santa Clara, CA, USA). In the presence of adaptor-dimers (electropherogram’s peak at 100 to 150-bp), another round of magnetic beads purification was performed. Libraries were quantified by both the Qubit^®^ Fluorometer (Life Technologies, Carlsbad, CA, USA) and the qPCR-based NEBNext library quantification kit (New England BioLabs, Hitchin, UK). Finally, equimolar amounts of each five index-tagged libraries were multiplexed together in one pool (2 pools, 10 single libraries) and then were sequenced by an Illumina HiSeq2500 for 50 sequencing cycles. The final design consisted of 5 libraries representing the GPO group (at T75) and 5 libraries representing the CTR group (at T75).

The raw 50 bp single-end sequences (Sanger/Illumina 1.9 encoding) were quality-controlled by FastQC v.0.11.4 (http://www.bioinformatics.babraham.ac.uk/projects/fastqc/), and the low-quality bases (quality scores < 20) and adaptor contamination (if present) were removed by Trimmomatic v.0.36 [17] using the parameters ‘ILLUMINACLIP:TruSeq3-SE:2:30:10 LEADING:3 SLIDINGWINDOW:4:15 MINLEN:25’. Quality assessment by FastQC was performed again on the clean reads to ensure contaminant-free data. The high-quality reads were mapped to the *Bos taurus* UMD3.1 genome assembly from Ensembl (https://goo.gl/QkbJJh) by HISAT2 v.2.1.5 [18]. The uniquely-mapped reads aligned to exons were counted with HTSeq v.0.6.1 [19], then tested by the DESeq2 R package v.1.14.1 [20] for the presence of differentially expressed genes (DEGs). The sequencing data (FASTQ files) associated with this study are deposited in the GenBank’s Sequence Read Archive (SRA) under the accession ID SRP144360.

To better visualize the gene expression patterns in the GPO and CTR samples, a principal component analysis (PCA) was used. The PCA plots were created using the Clustvis tool (https://biit.cs.ut.ee/clustvis/) by [21], and using the default singular value decomposition (SVD) imputation. The enriched gene ontology (GO) terms and signaling pathways associated with DEGs in GPO versus CTR groups were identified by the Gene Set Enrichment Analysis (GSEA, http://www.broadinstitute.org/gsea/index.jsp). GSEA is a computational method that identifies shared differential gene expression of predefined, functionally related gene sets representing biological pathways. This is quantified by using a different type of enrichment score, a weighted Kolmogorov-Smirnov-like statistic that evaluates if the members of the pathway are randomly distributed or found at the extremes (top or bottom) of the list [22]. All the Ensembl gene IDs were collapsed to their corresponding HUGO gene symbols, then the entire normalized transcriptome dataset was ranked by the logarithm transformed (base 2) fold-changes (FC), where the up- and down-regulated genes were assigned positive and negative values, respectively. The pre-ranked whole dataset was analyzed (1000 permutations) against the curated canonical KEGG pathways (c2.cp.kegg.v6.1), GO terms (c5.all.v6.1) and Hallmarks (h.all.v6.1) catalogs, from the Molecular Signatures Database (MsigDB) [22]. In the present study, all gene sets with a false discovery rate (FDR) ≤ 0.05 were considered as being the most negatively or positively (based on the normalized enrichment score; NES) enriched gene sets in the GPO samples.

### 2.6. IPA Analysis

Ingenuity^®^ Systems Pathway Analysis (IPA, Ingenuity Systems, Redwood City, CA, USA; http://www.ingenuity.com) was used to identify canonical pathways using as a dataset of 367 DEGs between GPO group and the CTR one (*p* < 0.05) recognized by using human and mouse orthologs (93% mapped). The significance of the canonical pathway was measured with the *p*-value and the ratio of DEG/number of genes in the pathway. IPA also allows upstream regulators prediction which expression is known to regulate the custom dataset; the significance is measured by an overlap *p*-value that measures whether there is an overlap between the dataset genes and the known target genes. Moreover, is also indicated the status (inhibition/activation) of the upstream regulator.

### 2.7. Protein-Protein Interaction Analysis (STRING)

We performed a protein-protein interaction network analysis using the genes involved in the cholesterol biosynthesis using STRING software (http://string-db.org/). We set the interaction score as 0.9, the highest permitted by the software to avoid false positive. The output is a figure was up to seven lines which predict different types of interaction are built.

### 2.8. Statistics

GraphPad Prism (GraphPad Software, La Jolla, CA, USA) was used for statistical analysis. Differences in cell subsets from peripheral blood and plasma samples were assessed using Student’s *t*-test while malondialdehyde (MDA) differences in meat samples was performed using ordinary two-way ANOVA.

## 3. Results

### 3.1. Effects of GPO-Supplemented Diet on Blood Biochemical Analysis

The complete blood cell count and major plasma biochemical parameters were measured to check for the health status of animals fed with the GPO-supplemented diet. Overall, no notable variations were recorded throughout the trial, and only the number of total white blood cells was slightly increased (*p* = 0.06) in the GPO-supplemented animals (Figure 1).

Likewise, plasma biomarkers showed no differences between the two groups. The only exception was glucose (*p* = 0.08), whose levels were lower in GPO group when compared to the CTR one (Figure 2).

### 3.2. Influence of GPO-Supplemented Diet on Calves’ Blood Transcriptome

We identified 367 genes whose expression was significantly different (*p* < 0.05) in the GPO versus CTR group (Appendix A). Of which, 167 and 200 DEGs were up- and downregulated, respectively. A principal component analysis (PCA), using either the full DEG list (Figure 3) or only the top 20 DEGs (Appendix A), was able to separate the GPO from the CTR group with the first 2 components (PC1 and PC2) amounting to 69% or 83.5% of the variation, respectively.

Next, to identify the enriched pathways and discover functionally-related gene functions, we used the Ingenuity Pathway Analysis software (IPA); Ingenuity Systems; (Qiagen, Inc., Valencia, CA, USA), which is known to be based on human orthologous. We noticed that most of the enriched pathways were those related to cholesterol biosynthesis (Table 3).

Similar results were obtained when using the GSEA software, which highlights functionally related genes that are regulated by similar conditions (Appendix A). Furthermore, a total of five genes involved in cholesterol biosynthesis pathway, namely Farnesyl-diphosphate Farnesyltransferase 1 (FDFT-1), Squalene Epoxidase (SQLE), NAD(P)-dependent Steroid Dehydrogenase-*like* (NSDHL), Methylsterol Monooxygenase (MSMO)-1, and Sterol-C5-desaturase (SC5D), were significantly (*p* < 0.05) downregulated. (Table 4, Figure 4). Finally, also the Low-Density Lipoprotein Receptor (LDLR), a member of the low-density lipoprotein receptor gene family, was proved to be significantly downregulated in the GPO group (Table 4).

We then performed protein-protein interaction analysis using genes involved in cholesterol biosynthesis using the STRING database [23]. Interestingly, all proteins appeared to interact each other (interaction score confidence 0.9), thus further validating the effects of a diet supplemented with GPO on cholesterol biosynthesis (Appendix A).

Worth mentioning, the Sterol Regulatory Element-binding Transcription Factors 1 and 2 (SREBF-1 and SREBF-2), which represent key transcription factors involved in biosynthesis of lipids and cholesterol, were predicted to be down-regulated (Table 5) but however only SREBF-1 was reported in our dataset (Appendix A).

This finding would additionally support the hypothesis that GPO potentially affects cholesterol (and lipid) metabolism, as mirrored by the decreased (*p* = 0.07) plasma cholesterol levels (Figure 5).

### 3.3. Effects of GPO-Supplemented Diet on Meat Cattle Lipid Oxidation

Besides the pathways related to cholesterol biosynthesis, lipid oxidation could also be affected by GPO supplementation restoring the redox homeostasis and preventing oxidative stress [24]. To confirm this relationship, we measured the time-dependent variation (T0 and T7) in malondialdehyde (MDA) levels in slaughtered carcasses kept at 4 °C. Carcasses from GPO group showed lower amounts of MDA compared to CTR, and such a decrease was statistically significant already at T0 and increased (*p* < 0.001) at T7 (*p* < 0.001) (Figure 6 and Appendix A).

## 4. Discussion

Driven by the lack of molecular data surrounding the effect of GPO supplementation in ruminants, the objective of this study was to profile the whole-blood transcriptome of GPO-supplemented calves, and to evaluate whether there is a GPO-induced transcriptomic signature. We decided to investigate peripheral blood since in previous nutrigenomics-based studies from our group, we detected with accuracy, differentially expressed genes [1,2]. Moreover, gene expression from this tissue reflects physiological and pathological events occurring in different tissues [25,26] Thus, the major finding of the present study was that a 10% dietary level of GPO can modulate the expression of considerable number of genes in calves and leave a transcriptomic signature that mainly reflects an antioxidant profile. To the best of our knowledge, this is the first whole-transcriptome (RNA-Seq) profiling of cattle fed with a GPO-supplemented diet.

Bioactive dietary components such as GPO have received great attention in the past decade. It is estimated that roughly 20% of the total weight of grape fruits used for wine and juice processing results in GPO, and this by-product, when left unused, poses a serious environmental and economic concern with regard to its storage, processing and disposal [10]. However, due to its low cost, high fiber content, and the presence of antioxidant polyphenols, GPO has been recently considered as an alternative feed ingredient for livestock, including ruminants [25,26]. However, data about the use of GPO-supplemented diets in ruminants are controversial [9,11,27]. On one hand, the most important limitation of the use of GPO as ruminant feed is the presence of a high amount of lignified fibers and tannins, with potentially negative effects on digestive nutrient utilization. On the other hand, these latter derivatives may improve the ruminal metabolism by increasing protein supply to the small intestine, accompanied by a decreased ruminal degradability and a decreased methanogenesis [28]. Several reports described the beneficial effects of GPO on animal product quality. For example, GPO may prevent damage of both unsaturated lipids of membranes (oxidative stress) and the meat/milk fatty acids profiles inhibiting oxidation [9,10,29].

Blood cell counts and biochemical plasma parameters as indicators of the general health status of animals were measured at the end of the experiment. Overall, all the parameters showed no statistically significant differences when compared to CTR. The slight increase in total leukocytes number might not qualify for a GPO-induced leukocytosis. Many physio-pathological conditions cause leukocytosis in cattle [30]; nevertheless, present and previously published data do not support the hypothesis of an adverse effect of GPO on total blood cell count.

From all the biochemical plasma parameters measured in the present study, only glucose and cholesterol blood concentrations (discussed later) were decreased (*p* < 0.08 and *p* < 0.07, respectively) in cattle fed with GPO-supplemented diet. Grape pomace polyphenols (in different formulations) have been shown to decrease blood glucose concentration in diabetic mice and rats; as a consequence, it has been hypothesized GPO might help regulate blood glucose levels [31,32]. Nevertheless, contradictory results have also been reported [33]. In cattle, previous studies showed no differences in blood glucose concentrations, as also found in the present study [10,34]. In any case, not only species-differences, but also the GPO polyphenolic content, its origin and dosage and the interactions with other feed ingredients should be taken into consideration when comparing the effects of dietary polyphenols on biochemical parameters [10].

In addition to their antioxidant activity, polyphenols have been shown to possess a number of cardioprotective and atheroprotective effects, including the reduction of plasma cholesterol levels. For instance, women consuming a lyophilized GPO powder or red wine showed a reduction in cholesterol LDL plasma concentration [35]. Also, GPO lowered plasma cholesterol concentration in humans [36]. The cholesterol biosynthesis pathway involves a total of twenty-five enzymatic steps [37], and our RNA-Seq data indicate that five genes involved in this pathway (FDFT1, SQLE, NSDHL, MSMO1, and SC5D) were downregulated in calves fed with a GPO-supplemented diet. The mevalonate pathway, otherwise known as 3-hydroxy-3-methylglutaryl-coenzyme A reductase (HMG-CoA reductase) pathway, is an important component of the hepatic cholesterol biosynthetic pathway. During the conversion of mevalonate to cholesterol, many important enzymes such as HMG-CoA reductase, FDFT1, and SQLE are involved in regulating the overall process [38]. Specifically, FDFT1 (squalene synthase) catalyzes the conversion of two molecules of farnesyl pyrophosphate to squalene in a two-stage reaction. Although HMG-CoA reductase is considered as the first rate-limiting enzyme of cholesterol biosynthesis (and the target for HMG-CoA reductase inhibitors such as statins), FDFT1 plays a key regulatory role; in fact, it is responsible for directing farnesyl pyrophosphate to either the sterol or non-sterol branch of the aforementioned pathway [39]. Once squalene is generated, SQLE catalyzes its epoxidation to 2,3-oxidosqualene (squalene epoxide). This is the first reaction of the cholesterol-specific biosynthetic pathway; furthermore, SQLE is considered as one rate-limiting enzyme of this same pathway [31,40]. In light of this, FDFT1 and SQLE are looked upon as perspective molecular targets for the development of novel lowering cholesterol drugs [41]. In our experimental conditions, GPO supplementation downregulated both genes, with a possible modulatory effect on cholesterol synthesis. Such an effect, potentially beneficial, was mirrored by the reduction of plasma cholesterol levels.

By contrast, few data are actually available about NSDHL and MSMO1, representing two other downregulated genes localized downstream the squalene epoxide synthesis. These transcripts have been associated with metabolic disease in humans [42,43]; however, the cattle NSDHL gene has been associated with transcriptomic changes occurring during the oocyte developmental competence [44], whereas MSMO1 (together with HMG-CoA reductase and other genes involved in cholesterol synthesis) was enriched in heifers fed with different forage-to-concentrate ratios [45]. Therefore, we hypothesize that the downregulation we observed might be due to a general rather than a gene-specific effect of GPO on cholesterol pathway. According to our expression data we predicted also the inhibition of SREBP 2 (*p*-value 3.39 × 10^−4^), one of the key transcription factor involved in the cholesterol biosynthesis further confirming so the influence of GPO on cholesterol biosynthesis.

Finally, it’s known that GPO contains high amount of polyphenols (e.g., anthocyanins, catechins, and flavonols) [26]. These bioactive derivatives have many health benefits, but their antioxidant properties are the most important. This antioxidant activity is exhibited through scavenging of free radicals and the inhibition of lipid oxidation [46]. Therefore, we measured MDA levels in slaughtered CTR and GPO cattle carcasses at T0 and kept at 4 °C for 7 days. Malondialdehyde is one of the biomarker most commonly used to measure oxidative damage [47]. Interestingly, our data showed a reduced amount of MDA in GPO carcasses immediately after slaughtering and after 7 days of conservation at +4 °C (*p* < 0.001). This result further suggests a beneficial effect (antioxidant) of GPO-supplemented diet and corroborates those previously obtained in other farm animals and showing that the administration of polyphenols from GPO reduced lipid peroxidation [48].

It should be noted that the concentrate that was given to the CTR and GPO groups in this study had a slightly different chemical composition. This was because we needed to re-calibrate the concentrate given to the GPO group to be isoenergetic and isoproteic after the inclusion of the 10% GPO; thus, changing its composition. This adds some limitation to the study, where some components of the CTR group’s concentrate contained an ingredient (e.g., dried orange pulp) that might have—per se—a slight antioxidant effect. However, the fact that GPO can induce a transcriptional signature that reflects an antioxidant activity—even in a non-antioxidant-deficient animals—should be valued.

## 5. Summary and Conclusions

This study presents the first RNA-Seq-based whole-transcriptome profiling of GPO-supplemented calves. The findings suggest that, in calves, 10% GPO supplementation leads to change of expression of several genes involved in the cholesterol and lipid metabolism pathways. In addition, this study provides evidence on the effects of GPO-supplemented diets supplemented on the blood-transcriptome signature in veal calves, which mainly reflects an antioxidant activity.

## Figures and Tables

**Figure 1 animals-08-00188-f001:**
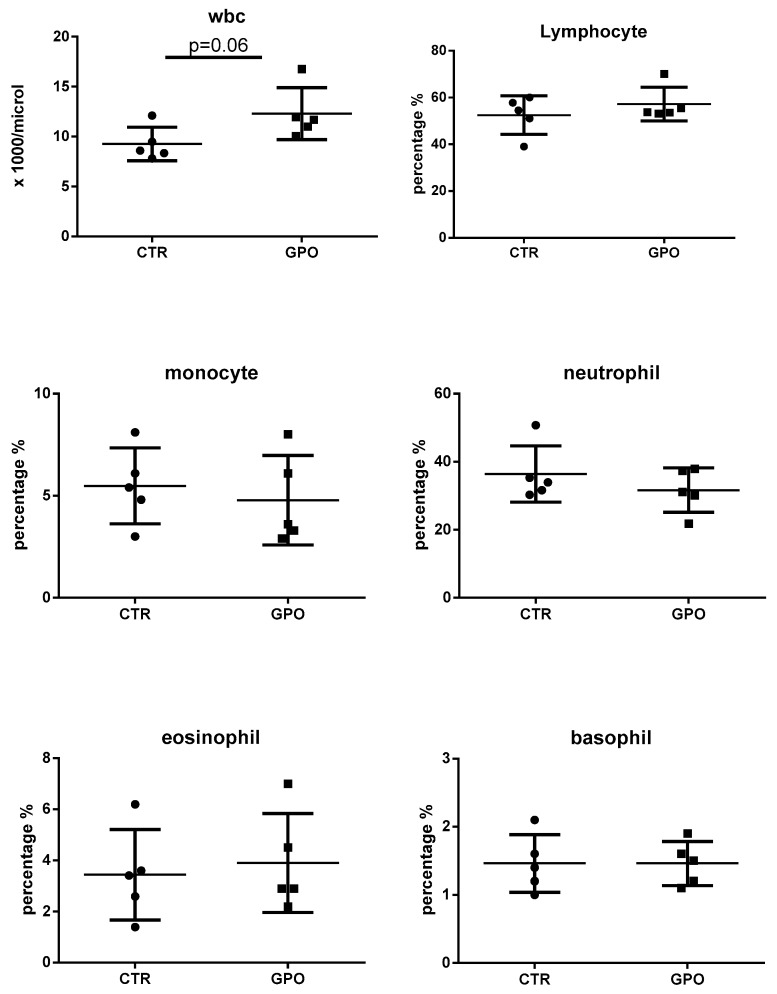
Effect of grape pomace (GPO)-supplemented diet on blood cell counts. Each point represents a single subject, and data are expressed as percentage. Any possible differences were analyzed by using the Student’s *t*-test. CTR: a control group.

**Figure 2 animals-08-00188-f002:**
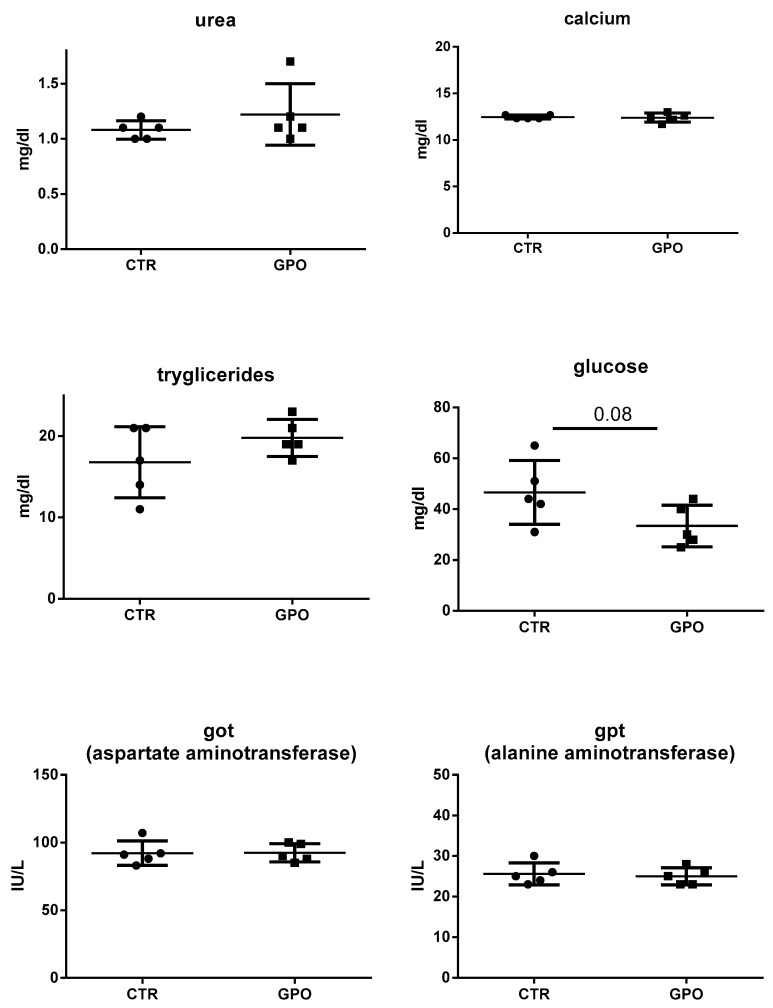
Effect of GPO-supplemented diet on selected plasma parameters. Each point represents a single subject, and possible differences were analyzed by using the Student’s *t*-test.

**Figure 3 animals-08-00188-f003:**
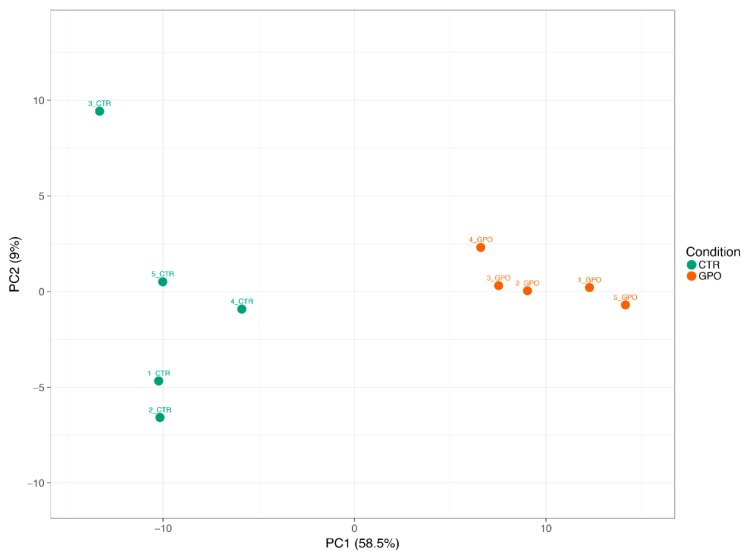
Principal component analysis (PCA) plot of the differentially expressed genes (DEGs) in veal calves after 75-days of grape pomace (GPO) supplementation, compared with unsupplemented control group (CTR). The GPO-supplemented animals (orange circles, right side) are separated from the unsupplemented CTR animals (green circles, left side) with the first two components (PCA1 and PCA2) accounting for 69% of the total variation. The PCA plots were created using the Clustvis tool (https://biit.cs.ut.ee/clustvis/) by [21]. For interpretation of the references to color in this figure legend, the reader is referred to the web version of this article.

**Figure 4 animals-08-00188-f004:**
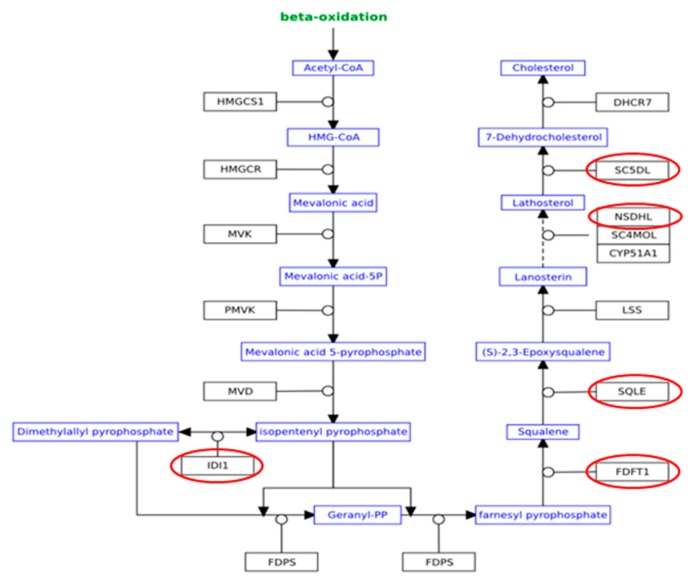
Cholesterol biosynthesis pathway. Downregulated DEGs (red ellipse) in veal calves fed with a GP-supplemented diet (*p* < 0.05). FDFT-1: Farnesyl-diphosphate Farnesyltransferase; SQLE: Squalene Epoxidase; NSDHL: NAD(P)-dependent Steroid Dehydrogenase-*like*; SC5D: Sterol-C5-desaturase; IDI-1: Isopentenyl-Diphosphate Delta Isomerase 1.

**Figure 5 animals-08-00188-f005:**
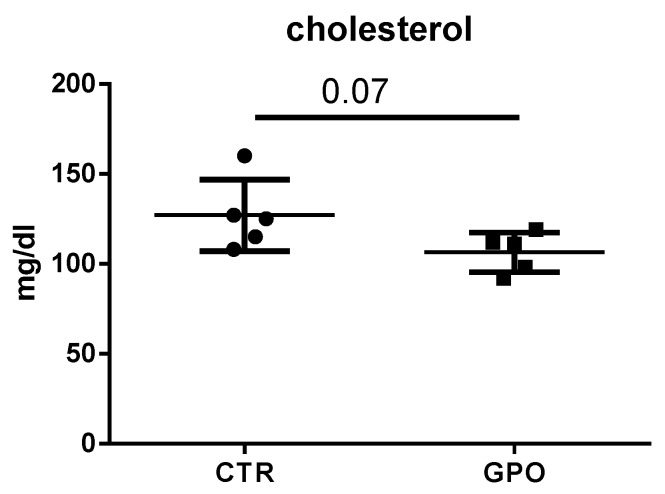
Effect of GPO-supplemented diet on plasma cholesterol levels. Each point represents a single subject and differences were analyzed using the Student’s *t*-test.

**Figure 6 animals-08-00188-f006:**
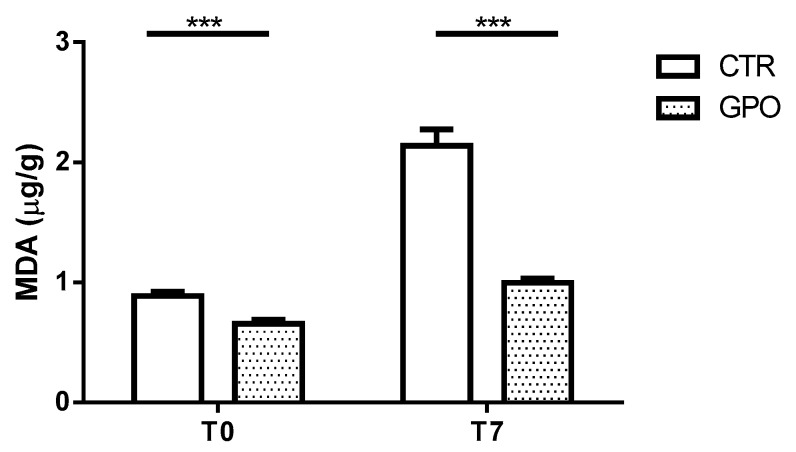
Malondialdehyde (MDA) levels at 0 (0.89 ± 0.03 vs. 0.65 ± 0.04) and 7 (2.14 ± 0.14 vs. 0.99 ± 0.03) days from slaughter in meat stored at 4 °C. Data represent mean ± SD, ***: *p* < 0.001, two-way ANOVA (5 samples/group).

**Table 1 animals-08-00188-t001:** Chemical composition of the alfalfa diet composition.

Parameters	%
Dry matter	87.13
Crude protein	16.35
Ash	9.47
Ether extract	1.21
Neutral detergent fiber	30.75
Acid detergent fiber	21.43
Acid detergent lignin	6.78

**Table 2 animals-08-00188-t002:** Ingredients and chemical composition of the custom-formulated diet.

Ingredient	Composition %	
	Control Group	Grape Pomace Group
Corn	22	34
Grain dust	14	22
Grape pomace flour	0	10
Fine bran	9.6	3.4
Sunflower seed	10	4.4
Faba bean	7.5	0
Biscuits waste	5.3	6.4
Distillers mais	5	1
Corn gluten feed	-	9
Barley	8.2	4
Bran wheat	5	0
Orange pulp dried	3.7	0
Soybean meal	2.9	0
Calcium carbonate	25	1.6
Molasses	1	1
Soybean hull	0.6	0
Common salt	0.6	0.6
Sodium bicarbonate 27%	0.5	0.8
Soybean oil	0.4	
Dicalcium phosphate	0	0.6
Magnesium oxide	0.4	0.4
Vitamin Premix	0.8	0.8
**Chemical composition of the concentrate**		
Dry matter	88.5	88.6
Crude protein	16.0	15.91
Ash	7.5	7.53
Ether extract	3.75	4.0
Crude Fiber	7.65	7.24
Starch	31.0	33.5
Neutral detergent fiber	20.37	18.90
Acid detergent fiber	7.95	8.32
Acid detergent lignin	3.50	3.86

**Table 3 animals-08-00188-t003:** Pathways represented among the DEG in GPO vs. CTR (Ingenuity Pathway (IPA) analysis).

Canonical Pathways	-Log (*p*-Value)	Ratio	Molecules
Cholesterol Biosynthesis I	5.76	0.385	FDFT1,SQLE,NSDHL,MSMO1,SC5D
Cholesterol Biosynthesis II (via 24,25-dihydrolanosterol)	5.76	0.385	FDFT1,SQLE,NSDHL,MSMO1,SC5D
Cholesterol Biosynthesis III (via Desmosterol)	5.76	0.385	FDFT1,SQLE,NSDHL,MSMO1,SC5D
Superpathway of Cholesterol Biosynthesis	5.25	0.222	FDFT1,SQLE,NSDHL,IDI1,MSMO1,SC5D
Epoxysqualene Biosynthesis	3.53	1	FDFT1,SQLE
Zymosterol Biosynthesis	2.37	0.333	NSDHL,MSMO1
IL-1 Signaling	2.33	0.0667	GNAI3,IL1A,PRKAR2B,MAPK8,GNA13,IRAK4
PPAR Signaling	2.26	0.0645	SRA1,IL1A,PDGFA,IL1RL1,PDGFRA,PTGS2
Toll-like Receptor Signaling	2.09	0.0694	UBD,IL1A,IL1RL1,MAPK8,IRAK4
Role of JAK family kinases in IL-6-type Cytokine Signaling	2.05	0.12	SOCS1,MAPK8,OSM
NF-κB Signaling	1.97	0.0462	IL1A,BCL10,MAPK8,FCER1G,PDGFRA,TBK1,MAP3K8,IRAK4
VDR/RXR Activation	1.95	0.0641	PDGFA,IL1RL1,HR,THBD,KLF4
Semaphorin Signaling in Neurons	1.94	0.0784	MET,RHOB,DPYSL3,PLXNB1
LXR/RXR Activation	1.91	0.0545	FDFT1,IL1A,LDLR,SREBF1,IL1RL1,PTGS2
Hepatic Fibrosis/Hepatic Stellate Cell Activation	1.88	0.0444	MET,IL1A,MYH9,PDGFA,IL1RL1,PDGFRA,SMAD7,COL18A1
Hepatic Cholestasis	1.78	0.0461	IL1A,PRKAR2B,SREBF1,IL1RL1,MAPK8,OSM,IRAK4
IL-8 Signaling	1.75	0.0421	GNAI3,MTOR,RHOB,MAPK8,PTGS2,GNA13,IRAK4,MYL12B
Nucleotide Excision Repair Pathway	1.72	0.0909	ERCC4,XPC,POLR2J

**Table 4 animals-08-00188-t004:** Gene expression values of cholesterol related genes.

Gene Symbol	Log2FC	*p*-Value
FDFT1	−0.31606	0.00267
SQLE	−0.2705	0.001057
NSDHL	−0.2293	0.016672
MSMO1	−0.26965	0.012593
SC5D	−0.27862	0.002411
SREBF-1	−0.16536	0.02396
LDLR	−0.37172	0.001701

**Table 5 animals-08-00188-t005:** The most activated and inhibited upstream regulators in GPO vs. CTR predicted by the Upstream Regulator Analysis in IPA.

Upstream Regulator	Molecule Type	Predicted State	*p*-Value
SREBF2	Transcription regulator	Inhibited	0.000339911
SIRT2	Transcription regulator	Inhibited	0.027754644
SREBF1	Transcription regulator	Inhibited	0.0015
CYP51A1	Enzyme	Activated	6.15 × 10^−5^
IL1B	Cytokine	Inhibited	1.15 × 10^−5^
POR	Enzyme	Activated	9.32 × 10^−5^
TLR9	Transmembrane receptor	Inhibited	0.00105
IL5	Cytokine	Inhibited	0.00315

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
