# Peer review of "RNA Sequencing-Based Whole-Transcriptome Analysis of Friesian Cattle Fed with Grape Pomace-Supplemented Diet"

_animals, 2018, doi:10.3390/ani8110188_

Round 1

Reviewer 1 Report

The paper by Iannacone et al., is overall well-written and has some merits , since the transcriptome of blood cells after treatment of cows with pomace has not been previously assessed and can provide important information on the effect of the treatment on the immune system. The paper present several serious limitations that need to be addressed:

-          A hypothesis is missing. Please, clearly provide a hypothesis. Also, in the introduction, the authors should provide some justification for the use of whole blood transcriptome.

-          Table 2. It is unclear why the two diets were so different. Some of the ingredients were altogether not present in the treatment group but only in the control. This is an issue. How the authors could test the effect of GP alone? It might be important to provide also a table with the chemical composition of the ratio, to see the differences in usual nutritional parameters; however, several of the ingredients might have antioxidant properties, for instance Orange pulp dried, that might provide confounding effect. It would have been better to keep the two diets the same and add the GP and an isoenergetic and isonitrogenous compound as supplement on top of the diet.

-          L159-160: this statement is very confusing. Did the authors used the FDR=0.05 or 0.01? Also, did the authors uploaded in GSE only the DEG or the whole dataset?

-          L163-166: please, provide more details about the ANOVA. What was the model used? Any random and fix effect?

-          L206-210: what tool was used to run the PCA? Please, provide details in the materials and methods. Why the PCA was used with the 20 most affected DEG (I assume the one with the lowest P-value)? It was probably better to use all the DEG.

-          L219-222: Please, provide details on the use of IPA in the materials and methods. What background was used to run IPA? This is an essential aspect that need to be reported when using an enrichment analysis tool (see Nucleic Acids Res. 2009 Jan;37(1):1-13) and the whole measure transcxriptome shoudl be used with IPA (i.e., uploaded dataset). Also, which cut-off was used for IPA (FDR=0.05 or 0.01)? Please, also clarify that only pathways analysis was performed using IPA.

-          L256: ibidem. The use of STRING software should to be described in details in Materials and Methods. Which dataset was used with STRING? Based on the comment it appears it was based on the genes of Table 4? If so, why not the whole dataset of DEG? A protein-protein interaction between the genes in Table 4 is not a novel finding but it is probably a given considering that they are part of the same metabolic pathway.

-          L265-269: this is confusing. The Table 5 report the results of the up-stream regulator IPA analysis while the authors indicated that the SREBF1 and 2 were down-regulated. (IPA provides an estimate of the activation/inhibition based on down-stream genes, in this case, also this is somewhat expected considering that the two transcription factors control cholesterol synthesis). Based on Table S1 only SREBF1 is down-regulated. As above, please, describe in Materials and Methods how the IPA was used.

-          L361-363: it is not a given fact that lower cholesterol is benefic for ruminants… Since this is one of the main conclusion of the paper it should be supported by previous literature or evidence.

-          L294-416: the discussion is excessively long and concentrated on few genes of the cholesterol pathways. The RNAseq dataset are meant to be analyzed in a systemic way. Focusing on few genes appear to defeat the original purpose of using RNAseq, as stated in the introduction. The authors have the benefit of having RNAseq data and some functional data (i.e., blood parameters). It would benefit more science if the discussion is done in a more holistic and systems biology way.

-          Conclusions. The conclusion is mostly a summary of the results. Maybe better to title “Summary and Conclusion”.  As pointed out above, I think the conclusion that GP improves the animal welfare because decrease cholesterol is not supported by any literature generated in ruminants. In human high cholesterol is associated with cardiovascular diseases that happens later in life. Here you have young calves; thus, it is probably irrelevant. Also, the authors should consider that most of the cholesterol produced in ruminants come from the liver; however,  the authors have found a decrease in cholesterol in immune cells (for the most part); thus, they should figure out why the cholesterol decrease in immune cells and what this mean.

Other minor points that need to be addressed:

-          L99: please, provide xg (or RCF) instead of rpm

-          L105-110: please, provide more details on which blood parameters were measured and with what kits

-          L150: could not access the deposited SRA dataset

-          L170-171: please, delete “ever” and “all”

-          Figure 2: the Y-axis labels are not readable. Not sure in which units the parameters are.

-          L186: please, provide a reference to support the fact that parameters were in the normal limit

-          L208: please, provide the whole dataset with the corresponding FDR so any scientist would be able to use your dataset to run enrichment analyses. It would be also better to have the dataset in an excel file.

-          Figure 7: the standard error appears excessively low….if so, I was wondering if the difference was not already present at T0.

Author Response

Reviewer #1

1.      Reviewer (R): The paper by Iannacone et al., is overall well-written and has some merits, since the transcriptome of blood cells after treatment of cows with pomace has not been previously assessed and can provide important information on the effect of the treatment on the immune system. The paper present several serious limitations that need to be addressed:

A hypothesis is missing. Please, clearly provide a hypothesis. Also, in the introduction, the authors should provide some justification for the use of whole blood transcriptome..

Authors (A): We thank the reviewer the appreciation of our work. We’ve accordingly revised the whole manuscript, including the summary (revision are tracked), the abstract, and the introduction (all changes are tracked). Also, our hypothesis has been now clearly stated in the introduction (Line 88-91).

We have used whole-blood because it is an accessible biological matrix that circulates the whole body and reflects the physio-pathological status of animals. Moreover, our previous nutrigenomics-based studies, using whole-blood, detected, with accuracy, differentially expressed genes (Elgendy et al., 2016; Elgendy et al., 2017) that were strongly correlated to qPCR cross-validation assays – using even other body tissues.

2.      R: Table 2. It is unclear why the two diets were so different. Some of the ingredients were altogether not present in the treatment group but only in the control. This is an issue. How the authors could test the effect of GP alone? It might be important to provide also a table with the chemical composition of the ratio, to see the differences in usual nutritional parameters; however, several of the ingredients might have antioxidant properties, for instance Orange pulp dried, that might provide confounding effect. It would have been better to keep the two diets the same and add the GP and an isoenergetic and isonitrogenous compound as supplement on top of the diet.

A: As the Reviewer highlights, we are aware that the small differences in the basal and the supplemented diets could be a limitation. However, using two different softwares for the data analysis (IPA and GSEA), we obtained similar results that were confirmed phenotypically by both cholesterol and lipid oxidation reduction. Still, we acknowledge, and explain, this limitation in the discussion section (Line 402-509).

3.      R: L159-160: this statement is very confusing. Did the authors used the FDR=0.05 or 0.01? Also, did the authors uploaded in GSE only the DEG or the whole dataset?

A: We thank the reviewer for the useful comment and we’ve now clearly stated that we used an FDR≤0.05 (Line: 192) and that we’ve used the whole dataset (Line 189). Moreover, we explained better how GSEA works (Line 179-186)

4.      R: - L163-166: please, provide more details about the ANOVA. What was the model used? Any random and fix effect?

A: We have used ordinary 2-way ANOVA without any post test. We are ready to provide raw data upon to the Reviewers/Readers, upon a reasonable request (Line 213)

5.      R: L206-210: what tool was used to run the PCA? Please, provide details in the materials and methods. Why the PCA was used with the 20 most affected DEG (I assume the one with the lowest P-value)? It was probably better to use all the DEG.

A: Initially we have used the PCA only with the 20 most affected DEG but upon reviewer’s suggestion, we now have performed PCA analysis with all DEGs and moved the previous PCA analysis as supplementary figure (Figure S1, Line 316). Finally, the legend of the new Figure 3 contains now the method used to generate the PCA (Line 327-329).

6.      R: -L219-222: Please, provide details on the use of IPA in the materials and methods. What background was used to run IPA? This is an essential aspect that need to be reported when using an enrichment analysis tool (see Nucleic Acids Res. 2009 Jan;37(1):1-13) and the whole measure transcriptome should be used with IPA (i.e., uploaded dataset). Also, which cut-off was used for IPA (FDR=0.05 or 0.01)? Please, also clarify that only pathways analysis was performed using IPA.

A: We thank the reviewer for the comment and we have included a new paragraph in the materials and methods section. In particular, as mentioned already in the first version, we ran IPA uploading all DEGs filtered with a p-value <0.05 and using human and mouse as background (Line 195-203).

7.      R: L256: ibidem. The use of STRING software should to be described in details in Materials and Methods. Which dataset was used with STRING? Based on the comment it appears it was based on the genes of Table 4? If so, why not the whole dataset of DEG? A protein-protein interaction between the genes in Table 4 is not a novel finding but it is probably a given considering that they are part of the same metabolic pathway.

A: We thank the reviewer for the comment and we confirm that we have used STRING software only uploading the genes involved in the cholesterol biosynthesis. We added a paragraph to the materials and methods section (Line: 205-209) explaining how the analysis were performed. Also, we’ve moved the figure to the supplementary material (Figure S3), as it seems to be redundant data as suggested by the reviewer.

8.      R:  L265-269: this is confusing. The Table 5 report the results of the up-stream regulator IPA analysis while the authors indicated that the SREBF1 and 2 were down-regulated. (IPA provides an estimate of the activation/inhibition based on down-stream genes, in this case, also this is somewhat expected considering that the two transcription factors control cholesterol synthesis). Based on Table S1 only SREBF1 is down-regulated. As above, please, describe in Materials and Methods how the IPA was used.

A: We tried to explain better the sentence clarifying that both transcription factor were predicted to be inhibited but, in our dataset, only SREBF1 was down-regulated (Line 383-384). Moreover, as requested by the Reviewer, we improved the materials and methods section also with the description of upstream regulators analysis (Line 200-203).

9.      R: L361-363: it is not a given fact that lower cholesterol is benefic for ruminants… Since this is one of the main conclusion of the paper it should be supported by previous literature or evidence.

A: We thank the Reviewer for the comment but we would like to keep the word ‘beneficial’ (Line 499). As we have mentioned in the discussion, GPO-supplemented diet has several positive effects and, in our study, we show a reduction of lipid oxidation which improves the meat quality and a reduction of cholesterol which could be potentially beneficial. Indeed, a reduced amount of cholesterol in the host has been associated with higher resistance to tuberculosis (https://doi.org/10.1073/pnas.0711159105). Because M.tuberculosis and M. bovis  share some similarities, it would be acceptable to say that modulation of cholesterol metabolism using GPO-supplemented diet, could potentially increase the resistance to intracellular pathogen like bovine tuberculosis in cattle.

10.  R: L294-416: the discussion is excessively long and concentrated on few genes of the cholesterol pathways. The RNAseq dataset are meant to be analyzed in a systemic way. Focusing on few genes appear to defeat the original purpose of using RNAseq, as stated in the introduction. The authors have the benefit of having RNAseq data and some functional data (i.e., blood parameters). It would benefit more science if the discussion is done in a more holistic and systems biology way.

A: Following the reviewer’s suggestion, thoroughly revised and shortened the whole discussion section (Line 421-510).

11.  R: Conclusions. The conclusion is mostly a summary of the results. Maybe better to title “Summary and Conclusion”.  As pointed out above, I think the conclusion that GP improves the animal welfare because decrease cholesterol is not supported by any literature generated in ruminants. In human high cholesterol is associated with cardiovascular diseases that happens later in life. Here you have young calves; thus, it is probably irrelevant. Also, the authors should consider that most of the cholesterol produced in ruminants come from the liver; however,  the authors have found a decrease in cholesterol in immune cells (for the most part); thus, they should figure out why the cholesterol decrease in immune cells and what this mean.

A: Following the reviewer’s suggestion, we thoroughly revised this whole section (Line 522-526).

Other minor points that need to be addressed.

12.  R  L99: please, provide xg (or RCF) instead of rpm

A: we changed 2000 rpm to 500 RCF (Line 126)

13.  R: L105-110: please, provide more details on which blood parameters were measured and with what kits

A: Following the reviewer’s suggestion, we provided more detail to the paragraph (Line 132-138)

14.  R: L150: could not access the deposited SRA dataset

A: The deposited SRA were made private until the publication of the article. If the reviewer wants to see, download or re-analyze the data we will be happy to provide an ftp link through which the data can be accessed. Also, the data will be publicly available from September 10th, 2018.

15.  R: L170-171: please, delete “ever” and “all”

A: We deleted “ever” and “all” (Line 218)

16.  R: Figure 2: the Y-axis labels are not readable. Not sure in which units the parameters are.

A: We apologize for the inconvenient and we made the Y-axis labels readable

17.  R: L186: please, provide a reference to support the fact that parameters were in the normal limit

A: We rephrased the sentence deleting the “normal limit” (Line 274)

18.  R: L208: please, provide the whole dataset with the corresponding FDR so any scientist would be able to use your dataset to run enrichment analyses. It would be also better to have the dataset in an excel file.

A: We thank the reviewer for the suggestion and already in the first version we have provided the full list of the 367 DEGs in an excel file with name, fold change (Log2FC) and significance (p-value).

19.   R: Figure 7: the standard error appears excessively low….if so, I was wondering if the difference was not already present at T0.

A: Indeed, samples were different already at T0 and this difference became significantly higher at T7 (Line 404). This finding further supports the positive effect of GPO supplementation on the quality of meat products. Moreover, raw data of the experiment are available upon request.

Reviewer 2 Report

The manuscript "RNA-Seq based whole transcriptome analysis of Friesian cattle fed with grape pomace-supplemented diet", is an interesting article about blood transcriptional profiles of cattle fed grape-pomace verses a control diet. Over all the science presented appears sound. The manuscript requires several small improvements in grammar which I have listed below. The manuscript also requires an update to the materials and methods section to improve upon and/or include the description of use of GSEA, IPA, and STING. This is also outlined below in order of appearance.

Line 85: What sex were the calves?

89: Change to - experiment, all calves received a standard basal diet for 120 days consisting of alfalfa hay plus a

91: Table 1 – Does this include the concentrate formulation? If not, can you report it in the manuscript? Or is it the same as the control diet in Table2?

92-96: Change to - After the 120 day acclimation period, the CTR group continued consuming this standard diet, while calves belonging to the GPO group received the same alfalfa hay as CTR group plus a concentrate containing 10% of GP flour (Table 2). The two diets were then fed for a total of 75 days, until the animals were slaughtered. At slaughter, the calves were 8.X +/- Y months of age and weighed 340 +/- 10 kg. (put in the actual mean and s.d. for age and weight).

98: Change to: Italia, Rome, Italy) during the slaughter process to measure..

99: Change to: After centrifugation at 2000 rpm for

101: Insert “the” between from and jugular

102: Change to: at two time-points; at the beginning (T0) and after 75

107: what is leukocyte formula?

108: Plasma samples were analyzed for what?

112: change to – meat samples

113: change till to until

119: Add “an: between traces, and in-column

121: Change processed to “until further processing”

138: Change to: (2 pools, 10 single libraries), and then were sequenced

151-152: Change to - DEGs in the GPO verses CTR groups were identified

152: To cite your use of the GSEA software, please reference Subramanian, Tamayo, et al. (2005, PNAS 102, 15545-15550) and Mootha, Lindgren, et al. (2003, Nat Genet 34, 267-273). Use reference #18 here.

151-162: Be more descriptive of what GSEA does - Given an a priori defined set of genes S (e.g., genes encoding products in a metabolic pathway, located in the same cytogenetic band, or sharing the same GO category), the goal of GSEA is to determine whether the members of S are randomly distributed throughout L (L= a ranked list according gene differential expression between the experimental classes) or primarily found at the top or bottom. We expect that sets related to the phenotypic distinction will tend to show the latter distribution.

170: delete “ever”

171: Delete “More in details,”

Figure 1: It is almost impossible to read. Axis titles and axes must be much larger.

186-187: Change to: were within normal limits, and no differences between CTR and GPO groups were observed.

187: delete “represented by”

188: Change if to when.

Figure 2: same comments as for Figure 1.

207: Change “affected” to “different”, delete “group”

209: Change to: verses the CTR group (Table S1).

210: amounting to 83.5% (of the variation?)

219-222: Use of IPA should be described in the MM.

219: Change to “pathways”

238: Change to “Similar results were obtained…”

255: Change 0,05 to 0.05

256: Change to “We then performed”

256: The input for STRING was the 7 genes? Clarify.

256-258: Use of STRING should be described in MM.

257: Change “were proved” to “appeared”

261: Change 0,9 to 0.9

265-267: Did you look for these transcription factors in your RNA-Seq dataset to verify the IPA prediction? If the use of IPA was described in the MM the discussion would be easier it follow as it seems this analysis comes from out of the blue.

257: Which results or pathways does the downregulation of LDLR support? Make this clear to help the reader connect the dots.

279: move “also” between could and be

278-293: Where was this procedure discussed in the MM?

295: change to “effects”

297: Delete e.g,

300: change tose to poses

301-304: Too many commas - Despite this, due to its low cost, high fiber content, and most of all the presence of antioxidant polyphenols, GP has been recently considered as an alternative feed ingredient for livestock, including ruminants

313 Change to ‘evidence strengthens”

314: Change to - GP as a by-product useful for more sustainable production, to improve the health status of cattle, and to provide beneficial effects for cattle meat and milk food-products.

334: Change to: Nevertheless, contradictory results have also been reported [28}.

342: Change to - Also, GP has been shown to lower plasma…

358: Change to - In light of this, FDFT1 and SQLE are looked upon as

361: What does “benefic” mean??? Beneficial?

362-3: Change to - even though it resulted in only a trend in our data.

366: Change to - however, the cattle

379: Change wherever to where

383: Change to - revealed that most genes coding for…

389-390: Change to - likewise other species [21,27,41–43]. Moreover, GP seems to have an affect at the transcriptional level upon the regulators (SREBPs) rather than then effectors, in particular those genes

413: Change to – suggests

418: Change to – alfalfa hay

418-419: Change to – on the blood transcriptome

420: Change to - of which 167 and 200 were up- and downregulated in GP, respectively.

Author Response

Reviewer #2

The manuscript "RNA-Seq based whole transcriptome analysis of Friesian cattle fed with grape pomace-supplemented diet", is an interesting article about blood transcriptional profiles of cattle fed grape-pomace verses a control diet. Over all the science presented appears sound. The manuscript requires several small improvements in grammar which I have listed below. The manuscript also requires an update to the materials and methods section to improve upon and/or include the description of use of GSEA, IPA, and STING. This is also outlined below in order of appearance.

1.      R: Line 85: What sex were the calves?

A: As kindly required by the reviewer, calves were all male and this has been stated now clearly in the manuscript (Line 103)

2.      R: Line 89: Change to - experiment, all calves received a standard basal diet for 120 days consisting of alfalfa hay plus a

A: We rephrased the paragraph following the reviewer’s suggestion, and now we hope that our revision sounds good (Line 103-110).

3.      R: 91: Table 1 – Does this include the concentrate formulation? If not, can you report it in the manuscript? Or is it the same as the control diet in Table2?

A: Yes, Table 1 includes the concentrate formulation which is the almost the same for the two diets reported in Table 2

4.      R: 92-96: Change to - After the 120 day acclimation period, the CTR group continued consuming this standard diet, while calves belonging to the GPO group received the same alfalfa hay as CTR group plus a concentrate containing 10% of GP flour (Table 2). The two diets were then fed for a total of 75 days, until the animals were slaughtered. At slaughter, the calves were 8.X +/- Y months of age and weighed 340 +/- 10 kg. (put in the actual mean and s.d. for age and weight).

A: As kindly requested by the Reviewer, actual mean and SD for age and weight were reported (Line 122)

5.      R: 98: Change to: Italia, Rome, Italy) during the slaughter process to measure.

A: We changed the sentence accordingly (Line 124)

6.      R: 99: Change to: After centrifugation at 2000 rpm for

A: We changed the sentence accordingly (Line 124)

7.      R: 101: Insert “the” between from and jugular

A: We changed the sentence accordingly (Line 128)

8.      R: 102: Change to: at two time-points; at the beginning (T0) and after 75

A: We changed the sentence accordingly (Line 128-129)

9.      R: 107: what is leukocyte formula?

A: The leukocyte formula is the percentage of different types of leukocyte cell measured in the peripheral blood. In our study we have measured lymphocyte, monocyte, basophil, eosinophils and neutrophils (Figure 1)

10.  R: 108: Plasma samples were analyzed for what?

A: Plasma sample were used for measuring urea, calcium, tryglicerides, glucose, AST, ALT as reported in fig 2

11.  R: 112: change to – meat samples

A: We corrected the word accordingly (Line 140)

12.  R: 113: change till to until

A: We added the word accordingly (Line 141)

13.  R: 119: Add “an: between traces, and in-column

A: We added the word accordingly (Line 147)

14.  R: 121: Change processed to “until further processing”

A: We changed the word accordingly (Line 149)

15.  R: 138: Change to: (2 pools, 10 single libraries), and then were sequenced

A: We revised the verb accordingly (Line 166)

16.  R: 151-152: Change to - DEGs in the GPO versus CTR groups were identified

A: We revised the sentence accordingly (Line 179-180)

17.  R: 152: To cite your use of the GSEA software, please reference Subramanian, Tamayo, et al. (2005, PNAS 102, 15545-15550) and Mootha, Lindgren, et al. (2003, Nat Genet 34, 267-273). Use reference #18 here.

A: The references list has been accordingly updated.

18.  R: 151-162: Be more descriptive of what GSEA does - Given an a priori defined set of genes S (e.g., genes encoding products in a metabolic pathway, located in the same cytogenetic band, or sharing the same GO category), the goal of GSEA is to determine whether the members of S are randomly distributed throughout L (L= a ranked list according gene differential expression between the experimental classes) or primarily found at the top or bottom. We expect that sets related to the phenotypic distinction will tend to show the latter distribution.

A: How GSEA works has been better described now (Line 181-194)

19.  R: 170: delete “ever

A: We deleted the word accordingly (Line 218)

20.  R: 171: Delete “More in details,”

A: We deleted the words accordingly (Line 218)

21.  R: Figure 1: It is almost impossible to read. Axis titles and axes must be much larger.

A: We apologize for the inconvenience and we made the figure more readable.

22.  R: 186-187: Change to: were within normal limits, and no differences between CTR and GPO groups

A: We rephrased the sentence, deleting the expression ‘normal limit’ (Line 274-276)

23.  R:187: delete “represented by”

A: We deleted the words accordingly (Line 275)

24.  R: 188: Change if to when.

A: We changed the word accordingly (Line 276)

25.  R: Figure 2: same comments as for Figure 1.

A: As before, we made Figure 2 more readable.

26.  R: 207: Change “affected” to “different”, delete “group”

A: We changed the sentence accordingly (Line 313-314)

27.  R: 209: Change to: verses the CTR group (Table S1).

A: We changed the sentence accordingly (Line 314)

28.  R: 210: amounting to 83.5% (of the variation?)

A: In the revised manuscript, we have replaced the PCA analysis figure with a new one, where we used all DEGs instead of the top 20 most deregulated genes. Finally, as requested by the reviewer, we specified that the percentage was referred to the variation (Line 315-317).

29.  R: 219-222: Use of IPA should be described in the MM.

A: A new paragraph which explains in detail the procedure used for IPA analysis has been added in the materials and methods section (Line 195-204).

30.  R: 219: Change to “pathways”

A: The word is been corrected accordingly (Line 330).

31.  R: 238: Change to “Similar results were obtained…”

A: We changed the sentence accordingly (Line 350).

32.  R: 255: Change 0,05 to 0.05

A: We changed the “,” with “.” (Line 372).

33.  R: 256: Change to “We then performed”

A: The paragraph is been fully revised (Line 377-380).

34.  R: 256: The input for STRING was the 7 genes? Clarify.

A: As specified in the previous point, the paragraph has been fully revised and Figure 5 now became Figure S3.

35.  R: 256-258: Use of STRING should be described in MM.

A: A paragraph about the analysis run using STRING is been added in the MM (Line 205-209).

36.  R: 257: Change “were proved” to “appeared”

A: This paragraph has been fully revised (Line 377-380).

37.  R: 261: Change 0,9 to 0.9

A: Figure 5 now became Figure S3 and the legend has been revised.

38.  R: 265-267: Did you look for these transcription factors in your RNA-Seq dataset to verify the IPA prediction? If the use of IPA was described in the MM the discussion would be easier it follow as it seems this analysis comes from out of the blue.

A: The paragraph about IPA which has been described in the materials and methods section. ALso, we specified in the manuscript that we have looked at these transcription factor in our DEGs list (table S1) and SREBF-1 was de facto down-regulated (Line 383-384).

39.  R: 257: Which results or pathways does the down-regulation of LDLR support? Make this clear to help the reader connect the dots.

A: As kindly requested by the reviewer, the role of LDLR has been specified in the manuscript (Line 356-359).

40.  R: 279: move “also” between could and be

A: We changed the sentence accordingly (Line 399).

41.  R: 278-293: Where was this procedure discussed in the MM?

A: The paragraph about the lipid oxidation measurement is now reported in the materials and methods section and the procedure was described citing a previous manuscript of our group (Line 139-144 and ref #17).

42.  R: 295: change to “effects”

A: a reply to that comment and all the following comments is provided at the end of this letter.

43.  R: 297: Delete e.g,

44.  R: 300: change tose to poses

45.  R:301-304: Too many commas - Despite this, due to its low cost, high fiber content, and most of all the presence of antioxidant polyphenols, GP has been recently considered as an alternative feed ingredient for livestock, including ruminants

46.  A: 313 Change to ‘evidence strengthens”

47.  R: 314: Change to - GP as a by-product useful for more sustainable production, to improve the health status of cattle, and to provide beneficial effects for cattle meat and milk food-products.

48.  R: 334: Change to: Nevertheless, contradictory results have also been reported [28}.

49.  R: 342: Change to - Also, GP has been shown to lower plasma…

50.  R: 358: Change to - In light of this, FDFT1 and SQLE are looked upon as

51.  R: 361: What does “benefic” mean??? Beneficial?

52.  R: 362-3: Change to - even though it resulted in only a trend in our data.

53.  R: 366: Change to - however, the cattle

54.  R: 379: Change wherever to where

55.  R: 383: Change to - revealed that most genes coding for…

56.  R: 389-390: Change to - likewise other species [21,27,41–43]. Moreover, GP seems to have an affect at the transcriptional level upon the regulators (SREBPs) rather than then effectors, in particular those genes

57.  R: 413: Change to – suggests

58.  R: 418: Change to – alfalfa hay

59.  R: 418-419: Change to – on the blood transcriptome

60.  R: 420: Change to - of which 167 and 200 were up- and downregulated in GP, respectively.

A: According to the Editor’s request, the discussion and the conclusion were shortened and thouroughly revised (without changing the main findings). For these reasons, the previous reviewer’s comments were not addressed  (not applicable anymore) because those parts have been deleted from the manuscript. We hope the new version encounters the flavour of the Reviewer.

Round 2

Reviewer 1 Report

I commend the authors for the effort to improve the manuscript. After the revision few issues still remain to be addressed:

-          It is still confusing the diet. How much basic diet (Alfalfa diet) and how much supplement was provided? Is the “custom-formulated concentrate” the same as the “custom-formulated diet”? The Table 2 is the composition of the whole ration or only the supplement? If only the supplement, please address L96-97. The caption of Table 2 indicate that also the chemical composition was reported, but the Table 2 only reports the ingredients and not the chemical composition.

-          Still the justification for the use of whole blood RNA is not provided. The response to reviewer provided by the authors can be acceptable but it is somewhat weak. They need to consider that they are reading the RNA of (mostly) circulating immune cells. Therefore, the interpretation of the data should be focused on these cells. While several parameters are the results of the metabolic and signaling activity of most tissues composing the body (as indicated previously the level of cholesterol is mostly driven by the liver) the RNA is quite specific and cannot be considered a systemic reading rather a cell-specific reading of activity. This is still missing from the paper and it is a fundamental aspect to consider during the discussion of the results that cannot be discarded. This is likely the major issue of this paper that need to be adequately addressed.

-          The authors should clearly indicate in the materials and methods which dataset they uploaded in IPA (they did so for GSE and STRING, thanks; however for STRING it is better to indicate that the DEG associated with cholesterol were uploaded, not all the genes associated with cholesterol). The use of the default background in IPA can be misleading because it is composed of all the genes annotated in human and mouse (it is possible in IPA to curtain this down by selecting the genes associated with various conditions/tissues; however, it still remain the fact that a good amount of genes in mouse and human are not available in bovine and viceversa. The best background would have been the use of the whole measured transcriptome. Either the authors change the analysis or clearly indicate this as a limitation of the study. It is likely that the results on the cholesterol would not change, but there might be surprises after the change of the background….

-          It is good that the authors added the limitation of the diet in the paper. However, the argument that the two different biological tools provided the same results would not support the goodness of the diet but the reliability of the used bioinformatics tools….

-          The comment on the statistical analysis was not addressed. The authors need to provide details on the statistical analysis model used indicating the fixed and random effect. OK not have done post-hoc in this case, since you have only 2 comparison and the FDR would be OK for it.

-          The authors did not address properly the comment about the PCA. They need to report in materials and methods (not in the caption of the figure, although they can keep it there as well) the tool and conditions used to run the PCA.

-          Despite the indication in the response to reviewer that the difference between IPA prediction and down-regulation of the two SREBF isoforms was discussed, I could not find such discussion in the manuscript. Just to comment on that. The activity and the gene expression do not always correspond, especially for transcription factors. SREBP needs to be activated (i.e., low cholesterol) or can be inhibited (i.e., CLA); thus, their activity can be quite independent from their gene expression. Please, address adequately this point in your discussion.

-          The claim that the low cholesterol is beneficial for the young ruminants has not been addressed. The paper indicated in the response to reviewer reports results in mice, not ruminants. It is important that the authors discuss the consequence of their finding (lower cholesterol) because this was the main finding of their paper. If the authors want to keep that the lower cholesterol is beneficial for the young calves they need to provide scientific support for it.

Author Response

Point by point reply to Rev 1

Reviewer (R): It is still confusing the diet. How much basic diet (Alfalfa diet) and how much supplement was provided? Is the “custom-formulated concentrate” the same as the “custom-formulated diet”? The Table 2 is the composition of the whole ration or only the supplement? If only the supplement, please address L96-97. The caption of Table 2 indicate that also the chemical composition was reported, but the Table 2 only reports the ingredients and not the chemical composition

Authors (A): we thank the reviewer for the comments. All animals received alfalfa diet and custom -ormulated supplement ad libitum until the slaughtering. In the first part of  table 2 we reported only the composition of the 2 different supplements (indeed the alfalfa hay is not shown)- while in the other table we reported the chemical composition. Moreover, the 2 custom-formulated concentrate have been calibrated to be isoenergetic and isoproteic as we already stated in the discussion during our first round revision (Lines: 103-107 and 411-415)).

R: Still the justification for the use of whole blood RNA is not provided. The response to reviewer provided by the authors can be acceptable but it is somewhat weak. They need to consider that they are reading the RNA of (mostly) circulating immune cells. Therefore, the interpretation of the data should be focused on these cells. While several parameters are the results of the metabolic and signaling activity of most tissues composing the body (as indicated previously the level of cholesterol is mostly driven by the liver) the RNA is quite specific and cannot be considered a systemic reading rather a cell-specific reading of activity. This is still missing from the paper and it is a fundamental aspect to consider during the discussion of the results that cannot be discarded. This is likely the major issue of this paper that need to be adequately addressed

A: We thank the reviewer for the comment and we have discussed the reasons which have driven our choice for using peripheral blood transcriptome analysis (Lines: 328-331)

R: The authors should clearly indicate in the materials and methods which dataset they uploaded in IPA (they did so for GSE and STRING, thanks; however for STRING it is better to indicate that the DEG associated with cholesterol were uploaded, not all the genes associated with cholesterol). The use of the default background in IPA can be misleading because it is composed of all the genes annotated in human and mouse (it is possible in IPA to curtain this down by selecting the genes associated with various conditions/tissues; however, it still remain the fact that a good amount of genes in mouse and human are not available in bovine and viceversa. The best background would have been the use of the whole measured transcriptome. Either the authors change the analysis or clearly indicate this as a limitation of the study. It is likely that the results on the cholesterol would not change, but there might be surprises after the change of the background….

A: It is true that some gene IDs could be not recognized by IPA (since we used human and mouse orthologs) but in our analysis the system mapped 93% of the entries and we believe that this value is good to go further with analysis. In the manuscript, line 200, we now specify this percentage.

R: It is good that the authors added the limitation of the diet in the paper. However, the argument that the two different biological tools provided the same results would not support the goodness of the diet but the reliability of the used bioinformatics tools…

A: We thank the Reviewer who acknowledge our efforts for explaining the limitations of our diet. However, as we described in the first revision round, the effects of GPO are “visible” even in a non-antioxidant-deficient animals which supports the overall effect of GPO on the whole blood transcriptome analysis.

R: The comment on the statistical analysis was not addressed. The authors need to provide details on the statistical analysis model used indicating the fixed and random effect. OK not have done post-hoc in this case, since you have only 2 comparison and the FDR would be OK for it

A: We thank the Reviewer for the comment and for improving the readability of the figure 6, we provided additional statistics information in table S2.

R: The authors did not address properly the comment about the PCA. They need to report in materials and methods (not in the caption of the figure, although they can keep it there as well) the tool and conditions used to run the PCA.

A: We thank the Reviewer for the comment and as suggested we added more information about PCA analysis in the ‘material and methods’ section (Lines: 181-183)

R: Despite the indication in the response to reviewer that the difference between IPA prediction and down-regulation of the two SREBF isoforms was discussed, I could not find such discussion in the manuscript. Just to comment on that. The activity and the gene expression do not always correspond, especially for transcription factors. SREBP needs to be activated (i.e., low cholesterol) or can be inhibited (i.e., CLA); thus, their activity can be quite independent from their gene expression. Please, address adequately this point in your discussion

A: Following the Reviewer’s suggestion, we commented on SREBF in the discussion section (Lines: 397-399)

R: The claim that the low cholesterol is beneficial for the young ruminants has not been addressed. The paper indicated in the response to reviewer reports results in mice, not ruminants. It is important that the authors discuss the consequence of their finding (lower cholesterol) because this was the main finding of their paper. If the authors want to keep that the lower cholesterol is beneficial for the young calves they need to provide scientific support for it.

A: We really appreciate the Reviewer comment and we tried to explain better why we think that low level of cholesterol could be beneficial even for young ruminants. As the Reviewer said in the comment, the manuscript cited in our previous reply is using mice as a model (https://doi.org/10.1073/pnas.0711159105) because mice are generally considered a good model for both M.tuberculosis and M.bovis infection. Moreover, other Authors have shown that M.bovis was shown able to metabolize cholesterol for sustaining its growth (doi10.1073pnas.0605728104). Besides bTB, host cholesterol is also important to sustain  Brucella abortus macrophage infection indicating a more general role for the cholesterol during the intracellular pathogen infectious disease (10.1128/IAI.70.9.4818-4825.2002). Thus, as indirect evidence, we believe that decreased cholesterol blood level could be beneficial against infectious disease and not only to prevent cardiovascular diseases.
